

# A predictive screening tool to detect diabetic retinopathy or macular edema in primary health care: construction, validation and implementation on a mobile application

Cesar Azrak[1,2], Antonio Palazón-Bru[2,3], Manuel Vicente Baeza-Díaz[1], David Manuel Folgado-De la Rosa[2], Carmen Hernández-Martínez[1], José Juan Martínez-Toldos[1] and Vicente Francisco Gil-Guillén[2,3]

[1] Ophthalmology Service, General Hospital of Elche, Elche, Alicante, Spain
[2] Department of Clinical Medicine, Miguel Hernández University, San Juan de Alicante, Alicante, Spain
[3] Research Unit, Elda Hospital, Elda, Alicante, Spain

## ABSTRACT

The most described techniques used to detect diabetic retinopathy and diabetic macular edema have to be interpreted correctly, such that a person not specialized in ophthalmology, as is usually the case of a primary care physician, may experience difficulties with their interpretation; therefore we constructed, validated and implemented as a mobile app a new tool to detect diabetic retinopathy or diabetic macular edema (DRDME) using simple objective variables. We undertook a cross-sectional, observational study of a sample of 142 eyes from Spanish diabetic patients suspected of having DRDME in 2012–2013. Our outcome was DRDME and the secondary variables were: type of diabetes, gender, age, glycated hemoglobin (HbA1c), foveal thickness and visual acuity (best corrected). The sample was divided into two parts: 80% to construct the tool and 20% to validate it. A binary logistic regression model was used to predict DRDME. The resulting model was transformed into a scoring system. The area under the ROC curve (AUC) was calculated and risk groups established. The tool was validated by calculating the AUC and comparing expected events with observed events. The construction sample ($n = 106$) had 35 DRDME (95% CI [24.1–42.0]), and the validation sample ($n = 36$) had 12 DRDME (95% CI [17.9–48.7]). Factors associated with DRDME were: HbA1c (per 1%) (OR = 1.36, 95% CI [0.93–1.98], $p = 0.113$), foveal thickness (per 1 μm) (OR = 1.03, 95% CI [1.01–1.04], $p < 0.001$) and visual acuity (per unit) (OR = 0.14, 95% CI [0.00–0.16], $p < 0.001$). AUC for the validation: 0.90 (95% CI [0.75–1.00], $p < 0.001$). No significant differences were found between the expected and the observed outcomes ($p = 0.422$). In conclusion, we constructed and validated a simple rapid tool to determine whether a diabetic patient suspected of having DRDME really has it. This tool has been implemented on a mobile app. Further validation studies are required in the general diabetic population.

Corresponding author
Antonio Palazón-Bru,
antonio.pb23@gmail.com

## INTRODUCTION

Diabetes mellitus is prevalent worldwide and its most important ophthalmological complications are diabetic retinopathy and diabetic macular edema (*Jeganathan, Wang & Wong, 2008*; *International Diabetes Federation, 2013*). The main consequences of these disorders are possible loss of vision and blindness (*Jeganathan, Wang & Wong, 2008*). Considering that these conditions can be treated in their initial stages it is important to diagnose them as early as possible and thus prevent the dire consequences (*Early Treatment Diabetic Retinopathy Study Research Group, 1991*; *Early Treatment Diabetic Retinopathy Study Research Group, 1995*).

The Valencian Community, an autonomous Mediterranean region in southeast Spain, has a population of some 5 million (*Instituto Nacional de Estadística, 2014*). A diabetic patient in this Community is mainly controlled by the primary care team, who refer the patient to the ophthalmological specialist when the patient reports sudden loss of visual acuity, is suspected of having macular edema, proliferative retinopathy, advanced retinopathy is shown on non-mydriatic retinography screening or there is evidence of cataracts (*Generalitat Valenciana: Conselleria de Sanitat, 2006*). This protocol is based on relevant scientific literature to treat diabetic retinopathy and maculopathy, for both primary health care and ophthalmology services. Thus, the diabetic patient has the diabetic retinopathy monitored independently of its severity. It is regularly managed by the primary care physician during its early stages and by ophthalmologists in more severe cases (*Generalitat Valenciana: Conselleria de Sanitat, 2006*).

The most described techniques used to detect diabetic retinopathy are non-mydriatic retinography and biomicroscopy of the retina (*Early Treatment Diabetic Retinopathy Study Research Group, 1991*; *Baeza et al., 2009*; *Ryan et al., 2013*), and those used to detect diabetic macular edema include biomicroscopy, angiography and optical coherence tomography (*Early Treatment Diabetic Retinopathy Study Research Group, 1995*; *Ryan et al., 2013*). However, most of these tests have to be interpreted correctly, such that a person not specialized in ophthalmology, as is usually the case of a primary care physician, may experience difficulties with their interpretation, as this is subjective. Accordingly, we undertook a study in the Valencian Community aimed at constructing and validating by means of a mathematical model a tool to detect diabetic retinopathy and diabetic macular edema using simple objective variables, which would therefore present no interpretation difficulties. In addition, as the mathematical model requires arithmetic operations, the tool has been implemented on a mobile app. The results of this study, therefore, provide a simple tool to help primary care services determine whether a diabetic patient needs to be referred to the ophthalmological specialist.

## METHODS

### Study population

The population study involved diabetic patients followed by the ophthalmological service of the General University Hospital of Elche (Valencian Community, Spain). These patients are referred from the primary health care services, according to the protocol (*Generalitat Valenciana: Conselleria de Sanitat, 2006*).

### Study design and participants

This cross-sectional, observational study was undertaken in a sample of diabetic patients referred to the ophthalmological service of the General University Hospital of Elche by the primary care teams between October 2012 and June 2013 and who were willing to participate. The sampling procedure consisted of randomly selecting one day each week (not always the same day) and recruiting all the diabetic patients who attended that day, by means of linear systematic sampling. A patient was considered to have diabetes if the diagnosis had been made by a physician (ICD9-MC 250.X). Patients were excluded if they had dementia, high myopia or another macular disorder, had had vitreoretinal surgery, a cataract operation during the previous 3 months, had received laser treatment in the macular area or panphotocoagulation, or were taking anti-angiogenic drugs.

### Variables and measurements

The main outcome variable was the presence of at least treatable diabetic retinopathy (severe, very severe or proliferating (*Ryan et al., 2013*)) or diabetic macular edema (DRDME). The diagnosis of these two disorders was made by clinical ophthalmological examination of the retina by indirect ophthalmoscopy and biomicroscopy of the central retina with a Topcon slit lamp, model SL-8Z (Topcon Corporation, Tokyo, Japan) using a 78 diopter lens (78D aspheric lens; Volk Optical Incorporated Company, Mentor, OH, USA) and indirect ophthalmoscopy with a 28D lens by an expert retinal ophthalmologist. Macular edema was defined as the presence of hard exudates or localized retinal thickening within a distance of 500 μm from the fovea, and the degree of diabetic retinopathy was defined according to the ETDRS study (*Early Treatment Diabetic Retinopathy Study Research Group, 1991*). The secondary variables were type of diabetes, gender, age (years), glycated hemoglobin (HbA1c) (%), foveal thickness (μm) and visual acuity (best corrected). The source of information for data concerning the type of diabetes, hypertension, dyslipidemia, gender, smoking, age and HbA1c was the clinical history. Best corrected visual acuity was obtained using the Snellen scale. The foveal thickness was obtained at the central area by dilating the pupil with a drop of tropicamide and measuring with spectral domain optical coherence tomography (Topcon 3D OCT 2000; Topcon Corporation®, Itabashi, Tokyo, Japan). The images were acquired by 512 horizontal linear scans and 128 vertical scans, centered on the fixation point making a 6 × 6 square 3D pattern. The mean retinal thickness was calculated automatically by the software of the device. We measured a 6 mm diameter area, centered on the fovea, thereby using for the study the central 1,000 μm area (the central circle).

## Sample size

The construction sample involved data from 106 patients, of which 35 had DRDME. To contrast an area under the ROC curve (AUC) different to 0.5, assuming a 95% confidence level and expecting to find an AUC of 0.9, the contrast power was nearly 100%. Using the same parameters in the validation sample (36 patients, 12 with DRDME), we obtained a contrast power of 97.17% (*Hanley & McNeil, 1982*).

## Statistical methods

The quantitative variables are reported as means and standard deviations, and the qualitative variables as absolute and relative frequencies. All the analyses were done with a significance of 5% and the confidence interval was calculated for each relevant parameter. The complete sample was divided into two parts. One part (80%) to construct the predictive model and the other part (20%) to validate the model constructed.

Construction: we used a binary logistic regression model to predict DRDME using the most clinically relevant variables, taking into account that we could only use a maximum number of explanatory variables in the model (10 observations of our outcome for each explanatory variable). The resulting model was transformed into a scoring system using the method of the Framingham study (*Sullivan, Massaro & D'Agostino, 2004*). The ROC curve was calculated and we constructed groups based on the probabilities of the model: low (<25%), medium (25–50%), high (50–75%) and very high (≥75%).

In the validation sample the AUC was calculated and the observed events compared with the expected events of the model using the $X^2$ test. No calculations were made of the classical indicators of a diagnostic test, such as sensitivity, specificity, predictive values and likelihood ratios, as the test constructed does not indicate a single value (positive or negative) but rather a probability of DRDME associated with each score. Accordingly, differences were studied between the expected (given by the predictive model) and the observed events to determine whether the reality corresponded to what was indicated by the model. On the other hand, calculating the AUC indicates the discriminating power of our model. A similar methodology has been used in other studies (*Palazón-Bru et al., 2015*).

## Mobile application

The models were implemented on a mobile application for the operating systems Android and iPhone (see Note S1). This application is free to download from any of the stores. Its name is *Diabetic retinopathy predictor*.

## Ethical consideration

The study was approved by the Ethics Committee of the General Hospital of Elche. All the patients signed the informed consent document. The study was undertaken in accordance with the basic principles of the World Medical Association Declaration of Helsinki and complied with the norms described in the European Union guidelines for good clinical practice.

**Table 1 Descriptive characteristics and analysis for diabetic retinopathy or macular edema in diabetic patients from a Spanish region.** 2012–2013 data.

| Variable | Construction sample $n = 106$ $n(\%)/x \pm s$ | Validation sample $n = 36$ $n(\%)/x \pm s$ | p-value | Adj. OR for DRDME (95% CI) | p-value |
|---|---|---|---|---|---|
| DRDME | 35(34.7) | 12(33.3) | 0.886 | N/A | N/A |
| DM type 2 | 90(85.7) | 26(76.5) | 0.207 | N/M | N/M |
| Female gender | 52(49.1) | 20(55.6) | 0.500 | N/M | N/M |
| Age (years) | 63.4 ± 14.4 | 62.8 ± 16.8 | 0.847 | N/M | N/M |
| HbA1c (%) | 7.7 ± 1.5 | 7.9 ± 1.8 | 0.643 | 1.36 (0.93–1.98) | 0.113 |
| Foveal thickness (μm) | 261.2 ± 71.3 | 285.2 ± 95.1 | 0.117 | 1.03 (1.01–1.04) | <0.001 |
| Visual acuity | 0.7 ± 0.3 | 0.7 ± 0.3 | 0.689 | 0.14 (0.00–0.16) | <0.001 |

**Notes.**

Abbreviations: Adj. OR, adjusted odds ratio; CI, confidence interval; DM, diabetes mellitus; DRDME, diabetic retinopathy or diabetic macular edema; N/A, not applicable; N/M, not in the model; $n(\%)$, absolute frequency (relative frequency); $x \pm s$, mean ± standard deviation.

Goodness-of-fit: likelihood ratio test = 53.4, $p < 0.001$; Nagelkerke $R^2 = 0.583$.

## RESULTS

Table 1 shows the information for the two samples. In the Construction sample ($n = 106$) there were 35 cases of DRDME (95% CI [24.1–42.0]) (retinopathy alone, 8; macular edema alone, 9; both disorders, 18). In the Validation sample ($n = 36$) there were 12 cases of DRDME (95% CI [17.9–48.7]), of which 6 had both disorders, 3 just macular edema and 3 just retinopathy. For the other characteristics, in both samples there was a majority of type 2 diabetes (76.5–85.7%), an older mean age (62.8–63.4 years), high HbA1c (7.7–7.9%), visual acuity of 0.7 and mean foveal thickness between 261.2 and 285.2 μm. No significant differences were found between the two samples ($p$: 0.117–0.886).

The factors associated with DRDME (Table 1) were: HbA1c (per 1%) (OR = 1.36, 95% CI [0.93–1.98], $p = 0.113$), foveal thickness (per 1 μm) (OR = 1.03, 95% CI [1.01–1.04], $p < 0.001$) and visual acuity (per unit) (OR = 0.14, 95% CI [0.00–0.16], $p < 0.001$). The scoring system with its risk groups created from these factors is shown in Fig. 1. In the model we selected the three most clinically relevant variables, as we had 35 outcomes in our sample (*Alasil et al., 2010*; *Buabbud, Al-latayfeh & Sun, 2010*; *Hermann et al., 2014*; *Varma et al., 2014*; *Palazón-Bru et al., 2015*).

In the Validation sample (Fig. 2) the AUC was 0.90 (95% CI [0.75–1.00], $p < 0.001$). No significant differences were found between the expected outcomes and the observed outcomes ($p = 0.422$) (Fig. 3).

## DISCUSSION

This study constructed an innovative tool able to determine whether a diabetic patient has diabetic retinopathy or diabetic macular edema, and who should therefore be referred to a specialist in ophthalmology. The model uses easily obtained variables and the calculation can be done easily and simply using a mobile application.

| Foveal thickness (μm) | Points |
|---|---|
| <200 | 0 |
| 200-249 | 1 |
| 250-299 | 2 |
| 300-349 | 3 |
| 350-399 | 4 |
| ≥400 | 5 |

| HbA1c (%) | Points |
|---|---|
| <8 | 0 |
| ≥8 | 1 |

| Visual acuity | Points |
|---|---|
| 0.9-1 | 0 |
| 0.6-0.8 | 1 |
| 0.3-0.5 | 2 |
| 0.0-0.2 | 3 |

| Points Sum | Risk (%) |
|---|---|
| Low (0-2) | 1.45-15.35 |
| Medium (3) | 39.15-39.20 |
| High (4) | 69.55-69.60 |
| Very high (≥5) | ≥89.05 |

**Figure 1** **Scoring system to predict diabetic retinopathy and diabetic macular edema.** HbA1c, glycated hemoglobin.

A literature search revealed no such multifactor models determining the likelihood of presenting DRDME, so that it is not possible to compare our model overall. However, we can make independent comparisons with the factors obtained. We found greater complications with higher levels of Hb1Ac and a greater foveal thickness, and a lower likelihood of DRDME as the visual acuity decreased. These particular findings are consistent with others (*Alasil et al., 2010*; *Buabbud, Al-latayfeh & Sun, 2010*; *Hermann et al., 2014*; *Varma et al., 2014*).

This study constructed and validated a tool to enable the physician to discriminate between those patients who should be referred to an ophthalmologist and those who do not need to be. Thus, the primary care physician should carry out regular control examinations to assess the possible need for referral. A patient found to be in the High or Very High group should be referred to the specialist, as the associated likelihood of DRDME is very high. On the other hand, if a patient is found to be in the Medium or Low groups (<3 points), the primary care physician should control the patient. A model such as the one explained herein generally presents certain difficulties in its use, as the model is usually complex and requires a certain time to do the calculation. However, our model uses very simple variables that, once introduced into the mobile application, give the likelihood of the outcome. Finally, a nursing professional who receives adequate training (use of the apparatus and measurement of visual acuity) could also use the proposed model and thus reduce the workload of the primary care physician.

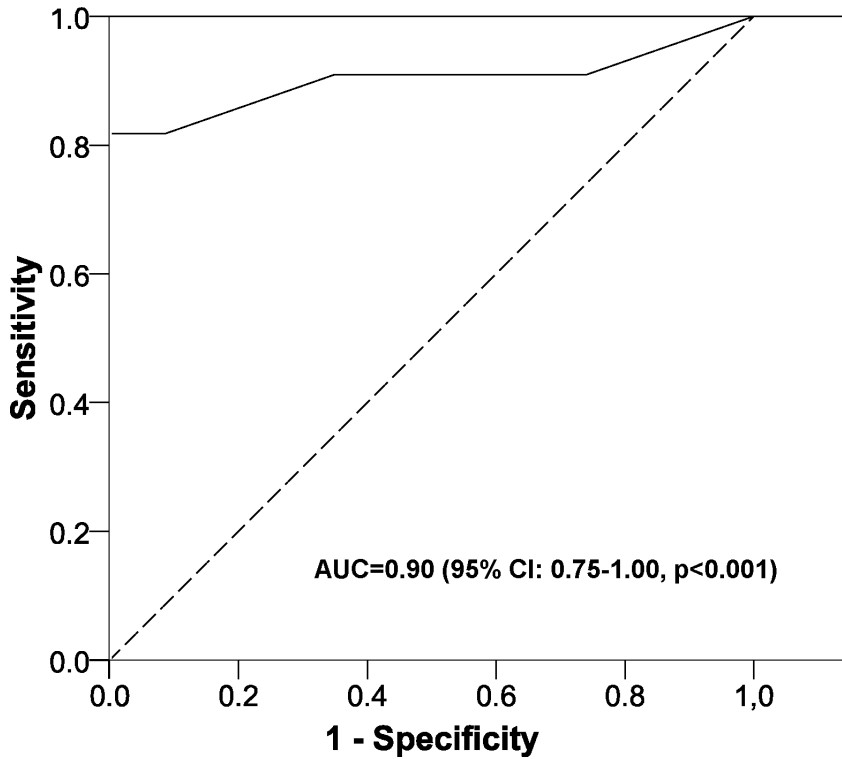

**Figure 2 Area under the ROC curve of the scoring system.** AUC, area under the ROC curve; CI, confidence interval.

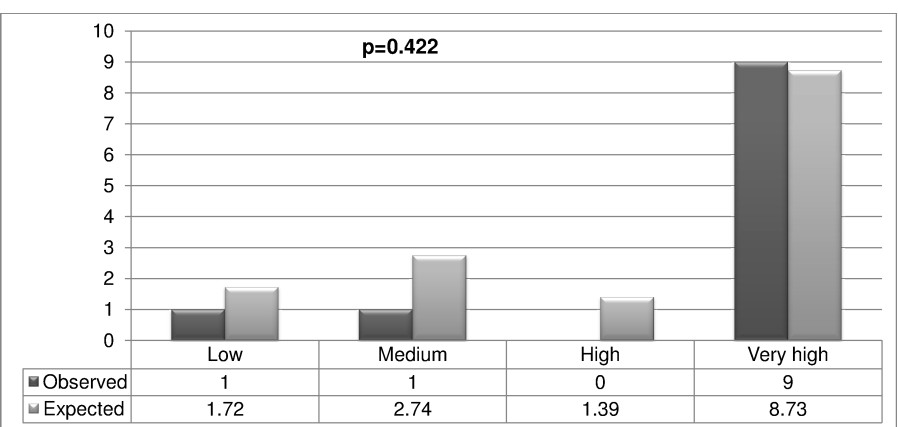

**Figure 3 Validation of the scoring system.**

   In some regions optical coherence tomography is not used in primary health care. Nevertheless, consideration could be given to its use in order to improve the screening of referable retinopathy. This tool, though, is expensive, so it is not feasible to include it at all health centers. However, an instrument could be purchased for each health care area (each area covers various health care centers), and diabetic patients suspected as having referable retinopathy could attend this particular center to undergo the test described here. This

method would reduce the possible cost considerably. Another possibility would be the use of a mobile unit that could rotate between the various health centers. This would help to integrate our predictive model into usual clinical practice, with the idea being to refer to the specialist services just those patients who really need it. Additionally, the inclusion of optical coherence tomography in primary health care could enable other macular disorders to be monitored, as this device can provide greater information about the macula. The resulting information, though, should be interpreted by an ophthalmologist, whereas the foveal thickness is a completely objective measure that can be interpreted with the other clinical parameters of our model by a healthcare professional working in the area of primary care. Finally, this study could be undertaken by other healthcare systems with different referral criteria, with the idea of reducing the proportion of patients who really need to be referred to the specialist ophthalmological services yet still following their own protocols.

### Strengths and limitations of the study

The main strength of this study concerns its novel way to construct and validate a scoring system to help primary care physicians take decisions about referring a diabetic patient to the ophthalmological specialist. Additionally, the factors used in the model are obtained objectively, so they do not have to be interpreted, as is the case with retinography.

Although the sample size may seem small, it is sufficient for the aims of the study (constructing and validating the model), as the contrast power was greater than 95%, whereas this is generally 80–90%. There are, too, non-significant variables in the model constructed. However, as in other studies, we have to consider that we are evaluating the model overall and not variable by variable, that is the goodness of fit (likelihood ratio test = 53.4, $p < 0.001$; Nagelkerke $R^2 = 0.583$) and the AUC in the validation sample (0.90) (*Palazón-Bru et al., 2015*). As this model was validated in diabetic patients who had already been referred to the specialist, further studies are needed in the general diabetic population to determine if it is applicable as a screening test for all diabetic patients when they attend their primary health care center (selection bias). These would study the probabilities of DRDME with our scoring system, as the general population would have a low prevalence of this macular disorder. On the other hand, concerning measurement bias, all the measurements were taken using calibrated devices and in accordance with current guidelines. Finally, we did not include other variables in the model that could have influenced DRDME, such as the duration of the disease. This particular variable was not assessed because the clinical history does not record it and the patient knows the duration of the disease only approximately. This could therefore cause an information bias, which is why it was not included in this study. Nevertheless, even without this variable the predictive model had an AUC of 0.90, which equates to a great discriminating power.

## CONCLUSION

This study constructed and validated a predictive model based on a scoring system to determine whether a diabetic patient referred to the specialist has DRDME. This model can be used to reduce the volume of patients referred to the ophthalmological services

from primary health care centers. Nevertheless, further studies should be undertaken to determine whether the model is applicable as a screening test in the general diabetic population.

## ACKNOWLEDGEMENT

We thank Ian Johnstone for help with the English language version of the manuscript.

### Funding

The authors received no funding for this work.

### Competing Interests

Antonio Palazón-Bru is an Academic Editor for PeerJ.

### Author Contributions

- Cesar Azrak conceived and designed the experiments, performed the experiments, wrote the paper, reviewed drafts of the paper.
- Antonio Palazón-Bru conceived and designed the experiments, analyzed the data, wrote the paper, prepared figures and/or tables, reviewed drafts of the paper.
- Manuel Vicente Baeza-Díaz conceived and designed the experiments, performed the experiments, reviewed drafts of the paper.
- David Manuel Folgado-De la Rosa conceived and designed the experiments, contributed reagents/materials/analysis tools, reviewed drafts of the paper and developed the mobile application.
- Carmen Hernández-Martínez and José Juan Martínez-Toldos conceived and designed the experiments, contributed reagents/materials/analysis tools, reviewed drafts of the paper.
- Vicente Francisco Gil-Guillén conceived and designed the experiments, reviewed drafts of the paper.

### Human Ethics

The following information was supplied relating to ethical approvals (i.e., approving body and any reference numbers):

The study was approved by the Ethics Committee of the General Hospital of Elche. All the patients signed informed consent. The study was undertaken in accordance with the basic principles of the World Medical Association Declaration of Helsinki and complied with the norms described in the European Union guidelines for good clinical practice.

### Supplemental Information

Supplemental information for this article can be found online at http://dx.doi.org/10.7717/peerj.1404#supplemental-information.

# PeerJ

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
