# Peer review of "A predictive screening tool to detect diabetic retinopathy or macular edema in primary health care: construction, validation and implementation on a mobile application"

_PeerJ, doi:10.7717/peerj.1404_

## Round 0.1 · original submission · Major Revisions

Both reviewers were conscientious and indicated constructive criticisms that would improve a revised manuscript. Please follow all of their advice as these reviewers will be asked to review a follow-up revision.

Reviewer 1 ·

Basic reporting

1. Overall, the article is well written. However, there are typos and grammatical mistakes that need to be fixed.
2. I believe the submitted article adheres to journal policies.
3. Figures are relevant to the content of the article.
4. The cited literature is appropriate and supports the body of the research.
5. The article includes sufficient introduction and background.

Experimental design

1. Although the authors claim that the purpose of the study is to create a feasible and applicable screening tool for primary care doctors, not all the factors included in this study are easy to obtain in regular primary care offices. One of the major factors that has been used to design and validate the algorithm is foveal thickness that is measured by OCT and demands high level of technical efficiency and availability of the device in the primary care settings. Despite the fact that foveal thickness may have a major role in detecting patients with macular edema, evaluation of this parameter by primary doctors or nursing staff may not be technically and financially possible.( OCT machine is very expensive). Please clarify if the authors think otherwise.
2. Evaluation of a patient with OCT is very useful and will provide detailed information about retina far beyond just foveal thickness. I believe performing an informative test such as OCT only for evaluation of the foveal thickness is counterintuitive when one can evaluate the whole retina structure during the same exam and therefore, perform an accurate assessment of the foveal structure and delineate any other retinal pathology.
3. Otherwise, I found the screening tool helpful for the primary care doctors when dealing with diabetic patients.

Validity of the findings

1. The study is statistically acceptable. However, I don’t totally agree with the inclusion criteria and the factors that have been used (specially foveal thickness) to design the screening algorithm (as I explained in previous section)
2. Additionally, the designed application has to be tested on a bigger group of diabetic patient in general population to avoid any bias (such as Hawthorne effect, etc …) associated to small group of patients who are already referred to a specialty clinic.
3. The authors may also be able to provide independent clinical data (retina specialist exam) to confirm that the patients who were detected by the application and marked as DRDME, actually have the disease. In another words, what is the false negative, false positive, positive and negative predictive value of the algorithm?

·

Basic reporting

Article has met these basic reporting standards.

Experimental design

The study states using the application in the primary health care setting, however it was validated in the setting of patients already referred to the ophthalmologist. These patients were therefore already suspicious and could lead to a selection bias for the validation of this tool.
The study also is essentially focused on finding advanced disease. It does not stratify from the original general medicine clinic those patients at risk for developing severe disease and potentially changing there course or referring to the ophthalmologist earlier. Macular edema and retinopathy respond best when treated early as chronic edema may not improve despite therapy. Therefore the app may get the patients to the ophthalmologist after it is too late.

Validity of the findings

A well-known metric for risk of retinopathy and edema is duration of disease from time of diagnosis. This should be considered as even diabetic patients under good/fair control (possibly A1c better than 8.0) can experience edema, retinopathy, or both over time.

Additional comments

Overall a well-written study. Its utility in this geographical setting may be applicable but it is unclear how this would translate to a different referral criteria model. In a geographical area that may have too many or too few referrals to the ophthalmologist for diabetic disease evaluation this tool could help streamline the thought process. However are patient’s with modifiable disease getting access early enough? The authors are able to validate their findings in this setting, however, and would provide benefit to the patients, doctors, and health system.

---

## Round 0.2 · accepted · Accept

One of the original reviewers returned to re-review your study, and they were happy with the revisions.

Reviewer 1 ·

Basic reporting

No comments

Experimental design

I agree with the rational behind the study design.

Validity of the findings

I believe the study is well designed and manuscript is well written and statistically acceptable.

Additional comments

I believe the supplementary information provided by the authors addressed my critics satisfactory. I will recommend the final draft with the most recent adjustments for publication.